# The Role of Passion in Self-Oriented Versus Team-Oriented Decision-Making in Team Sports

**DOI:** 10.3390/ijerph20032626

**Published:** 2023-02-01

**Authors:** Jany St-Cyr, Léandre Alexis Chénard-Poirier, Alexe Dufresne, Robert J. Vallerand

**Affiliations:** 1Laboratoire de Recherche sur le Comportement Social, Département de Psychologie, Université du Québec à Montréal, Montréal, QC H3C 3P8, Canada; 2Department of Management, HEC Montréal, Montréal, QC H3T 2A7, Canada

**Keywords:** obsessive passion, harmonious passion, decisions in sport, achievement goals

## Abstract

This study investigated the role of passion and achievement goals in making self-oriented and team-oriented decisions. Based on the Dualistic Model of Passion, it was hypothesized that in the context of collective sports, harmonious passion should lead to the adoption of mostly mastery goals, which in turn should lead to a more team-oriented decision-making. Conversely, obsessive passion should be related to the adoption of all three types of goals but mostly to performance-approach and performance-avoidance goals, which in turn should lead to a more self-oriented decision process. A total of 253 basketball players completed measures of passion and achievement goals in sport. They then were exposed to basketball scenarios and indicated their likelihood to act in a self-oriented or team-oriented manner. Results from structural equation modeling supported the hypotheses and lead to several implications for future research.

## 1. Introduction

Making decisions is an integral part of sport performance. For example, in a tight basketball game, deciding to force a shot or to make a pass to an open teammate can have a pivotal effect on the game result. Making the correct decision in a key situation may lead to victory, whereas making a poor decision may lead to defeat. Motivational factors can play a role in decision-making [1]. Passion for sport represents a motivational factor that characterizes most athletes [2]. As such, passion should play an important role in decision-making. However, not all passions are equal [3]. Some may lead one toward making the best decision irrespective of the situation, whereas other forms of passion may foster decisions that positively reflect on their self-esteem or their ego even though these decisions may lead to less-than-ideal outcomes for the team. The general goal of this study was to address this issue.

According to the Dualistic Model of Passion (DMP) [3,4,5], passion is a strong inclination towards an activity that people like or even love, find important, and in which they invest time and energy. This activity is so important for people that it becomes part of their identity. There are two types of passion: obsessive passion (OP) and harmonious passion (HP) [3,5]. HP is the result of an autonomous internalization of the activity in the identity of the person [3]. HP is linked to adaptive, mindful processes [6], which leads people to be fully involved in the activity, and thus to experience adaptive outcomes such as performance, positive emotions, and well-being [3,7]. Conversely, OP is the result of a more controlled internalization of the activity that people love [3]. More precisely, OP results from external or internal pressures aimed at maintaining or boosting people’s self-esteem, such as the desire to win at all costs and gaining social recognition [3,8]. In this case, the athletes’ ego is on the line, and they cannot fully engage in the activity. Thus, the process and outcomes are less than optimal, leading to internal and external conflict, negative emotions, avoidance coping, threat appraisals, and the adoption of risky behaviors [3,4,7].

Much less research has focused on the role of passion in cognitive processes. Such research has shown that HP leads to higher levels of flow, attention, and concentration than OP [3,7,9]. A cognitive process that has garnered much attention in psychology is cognitive goals, such as achievement goals [10,11]. Achievement goals are defined as goals for which the person’s primary aim is competence [12]. Thus, they are future-oriented cognitive representations that guide behavior toward an end state related to the skill that a person is committed to mastering, approaching, or avoiding [13]. Research on the role of passion in the use of achievement goals has shown that HP for a sport leads to more adaptive mastery goals (i.e., trying to master the activity), while OP has been found to be associated with mostly performance-approach (i.e., trying to do better than others) and performance-avoidance goals (i.e., trying to avoid doing worse than others) [14,15,16,17]. The relationship between OP and mastery goals is less clear since past research has shown positive [15,16,18] or non-significant [17,19] relationships between these variables.

In sports, cognitive processes such as decision-making also play an important role. The dynamic context of team sports involves episodes of high-intensity pressure and activity that require players to execute skills and make individual and collective decisions accurately and quickly to ensure success [20]. As highlighted by Ashford and colleagues [21], individual or collective decision-making can make the difference in a team’s performance between a win, a loss, or a draw. Only two studies have examined the role of passion in decision-making. In the first study, Schellenberg and Bailis [22] found that gamblers with an OP made worse decisions than those with a HP following various game outcomes. Similarly, in a second study with soccer referees, it was found that OP for refereeing in soccer led referees to engage in make-up calls (or favoring the team that was unjustly penalized in a prior decision) and thus making poorer decisions than those with HP [23] (Study 2).

In sum, research reveals that the role of athletes’ passion in decision-making process in team sports has yet to be assessed. Furthermore, the role of passion and how it may operate in decision-making situations wherein individual opportunities clash with team-oriented decisions has yet to be ascertained. For instance, if I have the choice between taking a shot to improve my scoring or making a pass to a teammate to ensure a team win, which option do I select and what are some of the associated psychological processes responsible for such a decision? The purpose of the present study was to address this issue. In line with the above, it was hypothesized that HP mostly predicts mastery goals, whereas OP predicts all three types of goals, but mostly performance-approach and performance-avoidance goals. Further, research has shown that mastery goals lead to collaborating with others [24,25] more than performance-approach and performance-avoidance goals [24,25]. It was thus hypothesized that HP would lead to team-oriented decisions whereas OP would mostly lead to self-oriented decisions because of their emphasis on performance-approach and performance-avoidance goals. Specifically, it was hypothesized that HP would relate positively to mastery goals, whereas OP would relate positively to performance-approach and performance-avoidance goals, and much less so to mastery goals. In turn, mastery goals should be positively associated with team-oriented decision-making and negatively to self-oriented decision-making. Performance-approach and performance-avoidance goals should be negatively associated with team-oriented decision-making and positively with self-oriented decision-making.

## 2. Materials and Methods

### 2.1. Participants and Procedure

A sample of 253 participants was recruited for this study on the Amazon Mechanical Turk website in exchange for a small monetary compensation. To participate in this study, participants had to have played basketball in an official team sanctioned by a league either at a high school, college, or professional level, and to still be playing basketball at the time of the study, either recreationally or in an organized league. All participants gave their informed consent prior to completing this study’s questionnaire. Participants were predominantly men (71.9%), with a mean age of 32.2 years (*SD =* 9.4) and were all living in the United States. Most participants played organized basketball in high school (64.0%), while others played either at a college (36.0%) or professional (2.8%) level, for an average of 8.0 (*SD* = 6.4) years of organized basketball.

### 2.2. Measures

#### 2.2.1. Passion toward Basketball

The Passion scale [5,26] was used to assess participants’ HP (6-item; α = 0.87; e.g., “Playing basketball is in harmony with the other activities in my life”) and OP (6-item; α = 0.92; e.g., “I have difficulties controlling my urge to play basketball”) toward basketball. Past research has consistently supported the validity of the Passion Scale across numerous activities, including sports [2,27]. Participants were asked to answer each item using a Likert scale ranging from 1 (“Not agree at all”) to 7 (“Very strongly agree”).

#### 2.2.2. Achievement Goals

Achievement goals were measured using Elliot and Church’s scale [28] that was adapted to basketball [16] (Study 2). Participants were informed that the items were related to their thoughts and feelings toward basketball. Four items for each of the three subscales were used to measure mastery (α = 0.88; e.g., “I want to develop my skills as much as possible”), performance-approach (α = 0.88; e.g., “I am motivated by the thought of outperforming my teammates”), and performance-avoidance goals (α = 0.85; e.g., “I just want to avoid playing poorly”). Participants were asked to assess their achievement goals using a Likert scale ranging from 1 (“Not agree at all”) to 7 (“Very strongly agree”).

#### 2.2.3. Self-Oriented and Team-Oriented Decision-Making

To assess basketball players’ self-oriented and team-oriented decision-making processes, real-life game scenarios were created for this study (see Appendix A for the scenarios). For each scenario, a general game situation statement was first presented to participants. There were two types of scenarios. In the first one, the outcome of the game was at stake and the decision had a direct effect on it (e.g., “There are 5 s remaining in the game, you team is down by 1 point, and you have the ball. What are the chances that you will…”). In the second type of scenario, the result of the game was not at stake so the decision would have no direct effect on it. Then a course of action or decision was presented. One course of action was self-oriented (e.g., “Take a difficult shot, but if you miss, it’s a guaranteed Fastbreak for the other team”). The other course of action was team-oriented (e.g., “Pass the ball to an open teammate in front of you”). Participants had to indicate how likely they would act according to each type of decision using a scale ranging from 0 (“Not likely at all”) to 10 (“Extremely likely”), with 5 being a “Neutral” choice.

The scenarios and their responses were first created by one of the authors who was, at the time of this study, a varsity basketball player. Scenarios were then reviewed by three experts in basketball and sports psychology, including a former national-level basketball player. Experts evaluated each scenario based on their ecological validity (i.e., the extent to which they represented a realistic game situation) and whether each course of action depicted a game decision that would either promote a self versus team orientation and influence the outcome of the game or not. Results from a confirmatory factor analysis (CFA) supported the 4-factor structure (Self- vs. Team-oriented decision that influences or not the game outcome) and the empirical distinctiveness of each type of decision. Specifically, the analysis showed excellent fit indices (χ^2^ [48] = 41.38, *p* = 0.74; CFI = 1.00; TLI = 1.00; RMSEA = 0.00, 90% CI [0.00–0.03]; SRMR = 0.04), along with good factor loadings and internal consistency (Self-oriented decisions that influence the outcome: 0.79 < λ < 0.81, α = 0.84, *M* = 5.61, *SD* = 2.66; Self-oriented decisions that do not influence the outcome: 0.54 < λ < 0.75, α = 0.67, *M* = 7.40, *SD* = 1.79; Team-oriented decisions that influence the outcome: 0.71 < λ < 0.84; α = 0.82, *M* = 8.30, *SD* = 2.02; Team-oriented decisions that do not influence the outcome: 0.63 < λ < 0.73, α = 0.73, *M* = 8.20, *SD* = 1.87). Considering the sample size and the number of variables in this study, the means of the four factors were calculated. Then, two latent variables were created to represent self-oriented and team-oriented decisions, each composed of both types of scenarios (i.e., those that influence the outcome of the game and those that do not). This allowed the model under study to have sufficient statistical power and converge properly [29].

## 3. Results

Analyses were conducted on Mplus 8.6 [30] and relied on robust maximum likelihood estimation (MLR). Variables’ means, standard deviations, and Pearson correlations based on scale scores are presented in Table 1. The following criteria were used to assess the adequacy of the model: a non-significant χ^2^, a CFI and TLI greater than or equal to 0.90, a RMSEA less than or equal to 0.08 with a 90% confidence interval between 0.00 and 0.10, and a SRMR less than or equal to 0.08 [31,32]. Results from a CFA including the seven variables measured in the model supported the statistical adequacy of the measurement model (χ^2^ [502] = 897.24, *p* < 0.001; CFI = 0.91; TLI = 0.90; RMSEA = 0.06, 90% CI [0.05–0.06]; SRMR = 0.08). Latent correlations among variables estimated by this CFA are shown above the diagonal in Table 1.

To test the hypothesized structural model, the measurement model was transformed into a nested mediation model using structural equation modeling (SEM) [33]. First, a full SEM model was tested in which HP and OP were set as predictors of mastery, performance-approach, and performance-avoidance goals. Then, each type of achievement goals was set as a predictor of self-oriented and team-oriented decisions. Covariance paths were also added between HP and OP [3] and among the three achievement goals, as research has shown that achievement goals are not mutually exclusive and individuals may adopt more than one type of goal for a specific activity [34]. This model did not yield an adequate fit to the data (χ^2^ [506] = 929.09, *p* < 0.001; CFI = 0.90; TLI = 0.89; RMSEA = 0.06, 90% CI [0.05–0.06]; SRMR = 0.08). Since more parsimonious models tend to have a better fit to the data [33,35], non-significant paths were removed. In addition, as suggested by modification indices, a direct path from OP to self-oriented decisions was added. This second more parsimonious model was retained since it showed an adequate fit to the data (χ^2^ [509] = 909.45, *p* < 0.001; CFI = 0.91; TLI = 0.90; RMSEA = 0.06, 90% CI [0.05–0.06]; SRMR = 0.08).

The standardized beta coefficients of the full SEM model are presented in Figure 1. Results showed that HP was positively related to mastery (b = 1.08, β = 0.86, *p* < 0.001, *SE* = 0.08) and performance-approach goals (b = 0.41, β = 0.25, *p* = 0.001, *SE* = 0.08). OP was positively related to performance-approach (b = 0.45, β = 0.49, *p* < 0.001, *SE* = 0.07) and performance-avoidance goals (b = 0.39, β = 0.60, *p* < 0.001, *SE* = 0.06) while being negatively related to mastery goals (b = −0.20, β = −0.29, *p* = 0.001, *SE* = 0.08). In turn, mastery goals were positively related to team-oriented decisions (b = 1.08, β = 0.67, *p* < 0.001, *SE* = 0.08), whereas performance-approach goals were negatively related to team-oriented decisions (b = −0.39, β = −0.32, *p* < 0.01, *SE* = 0.12). Finally, performance-avoidance goals (b = 0.71, β = 0.31, *p* < 0.001, *SE* = 0.08) as well as OP (b = 0.70, β = 0.47, *p* < 0.001, *SE* = 0.08) were both positively related to self-oriented decisions.

The significance of the indirect effect was tested using a 95% bias-corrected bootstrap confidence interval, based on 5000 resamples [36]. As bootstrapped confidence intervals are not available in MLR, they were estimated using maximum likelihood (ML). All other estimates were obtained using MLR. Note that MLR provides standard errors and fit indices that are robust to non-normality. It only affects standard errors, but not parameter estimates. Results showed that mastery goals mediated the relationship between HP (b = 0.57, *p* < 0.001, *SE* = 0.12, 95% [0.34–0.78]) and team-oriented decisions. Mastery goals also mediated the relationship between OP and team-oriented decisions (b = −0.19, *p* < 0.01, *SE* = 0.07, 95% [−0.35–−0.08]). In addition, performance-approach goals mediated the relationship between HP and team-oriented decisions (b = −0.08, *p* < 0.05, *SE* = 0.04, 95% [−0.19–−0.02]) as well as between OP and team-oriented decisions (b = −0.16, *p* < 0.05, *SE* = 0.06, 95% [−0.27–−0.02]). Performance-avoidance goals mediated the relationship between OP and self-oriented decisions (b = 0.18, *p* = 0.001, *SE* = 0.06, 95% [0.09–0.32]).

## 4. Discussion

The overall goal of this study was to explore the role of passion in self- versus team-oriented decisions. It was hypothesized that HP would be positively related to mastery goals that, in turn, would positively predict selecting team-oriented decisions and negatively predict selecting self-oriented decisions. Conversely, it was hypothesized that OP would positively predict performance-approach and performance-avoidance goals and much less so mastery goals. In turn, mastery goals were expected to positively predict self-oriented decisions and to negatively predict team-oriented decisions. Overall, the results generally supported the hypotheses and lead to some important implications.

In line with previous research, our results suggest that HP promotes a balanced road throughout sport engagement, allowing one to fully develop as an individual within a team environment [3,15,16,18]. In general, HP allows athletes to adopt mastery goals leading them to fully embrace the team orientation and a collaborative approach [24,25]. In other words, focusing on personal growth allows athletes to prioritize their team in their decision-making process regardless of whether the outcome of the game is at stake. This is in line with other research that showed that mastery goals, as promoted by HP, lead to more adaptive consequences, such as higher levels of team performance [16,17]. However, it should be noted that, in line with Verner-Filion and colleagues [17] (Study 2), HP was also positively associated with performance-approach goals that were negatively related to team-oriented decisions. One possible explanation for this result is based on the sport context in which competition promotes personal performance and social comparisons. Thus, overall, adopting a mastery approach appears to be a more suitable way to engage in team sport as it mainly promotes mastery goals, although it also predicts performance-approach goals.

Results also showed that OP was negatively associated with mastery goals and positively with performance-approach and performance-avoidance goals. The pathways from OP to mastery and performance-avoidance goals would appear to be suboptimal as such goals led to the decision to act in selfish ways and to act against the team’s best interest. These findings are in line with past research that showed that the adoption of performance, and especially performance-avoidance, goals leads to maladaptive behaviors [15,16,17]. Indeed, setting performance-approach and performance-avoidance goals leads athletes to be more concerned with their own performance and social comparison [28], which may lead to wanting to outdo others or avoiding performing less well than their teammates [17], thereby undercutting team performance. Furthermore, this more individualistic approach [24,25] leads athletes to prioritize their own success in their decision-making. When individual and team performance clash, the former will be selected when one’s passion is obsessive. In light of past research, it can be argued that such selfish decisions may be partly responsible for OP leading to athletes’ development of poorer relationships with their teammates [14,37]. In addition, OP was directly and positively related to self-oriented decisions. With OP, one’s identity is contingent on the activity such that decisions are made to protect one’s self-esteem or to benefit personally [3]. Therefore, unlike HP, OP appears to foster making decisions that are oriented toward personal rather than team benefits and that, in the end, are not optimal to team success.

The present findings lead to many practical applications. Indeed, results highlight the importance of fostering HP in sport and how it can benefit a team-oriented decision. First, HP can be fostered through the autonomy support that coaches offer to athletes. The DMP states that autonomy support facilitates the development and maintenance of HP, whereas controlling behaviors facilitate OP [3]. Coaches can support athletes’ autonomy in many ways, such as offering choices, providing task rationale, avoiding punishment, providing non-controlling feedback, and recognizing athletes’ feelings and perspectives [38]. Second, harmonious engagement in sport can also be facilitated by promoting athletes’ life balance. Balancing sport with other aspects of their lives (e.g., school, work, family) can allow athletes to flexibly engage in their sport, ultimately leading to positive consequences not only in sport, but also in other aspects of their life [3]. Third, it has been shown that increasing the use of signature strengths is related to higher levels of HP and well-being [39]. Thus, coaches could guide athletes in the discovery of their own athletic strengths and lead them to use these strengths in sport. In doing so, athletes should feel that they are using their full potential and engage more harmoniously in their sport, thereby leading to team-oriented decisions.

A few limitations warrant attention. First, this study relied on the use of self-reporting measures. Future research should replicate the present findings with observational methods. Second, the study was not experimental. Therefore, any inference of causality is precluded. Future research should use experimental designs and randomly assign athletes to either HP or OP induction (as in [40,41]) and assess whether it leads to the same findings as in the present study. Third, only intentions of decision-making were assessed. Future research should replicate these findings in real in-game situations.

## 5. Conclusions

In summary, the present study sheds light on the influence of passion in decision-making processes in a team sport. Through different relationships with achievement goals, HP and OP lead athletes to respectively make team versus individualistic decisions. Future research is necessary to pursue this study and determine if passion can also influence other types of decision-making processes.

## Figures and Tables

**Figure 1 ijerph-20-02626-f001:**
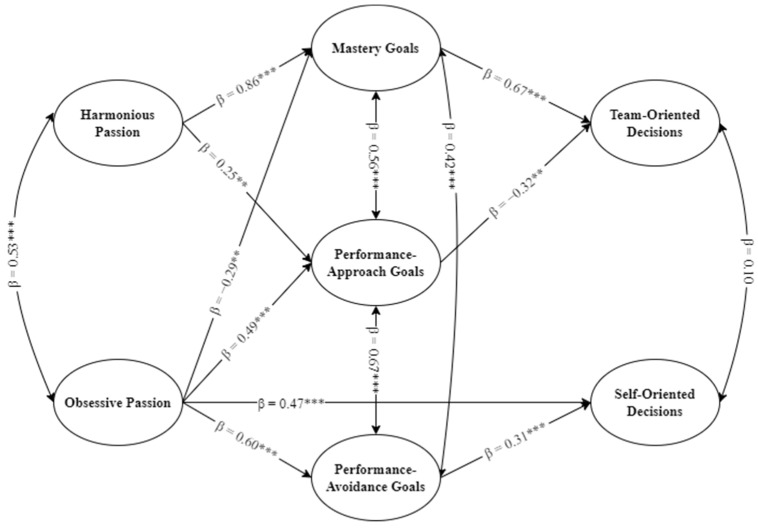
Structural Equation Model-Relations Between Passion, Achievement Goals, and Decisions. *N* = 253. β = standardized beta. ** *p* < 0.01; *** *p* < 0.001.

**Table 1 ijerph-20-02626-t001:** Means, Standard Deviation, and Correlations among Model Variables.

	*M*	*SD*	1.	2.	3.	4.	5.	6.	7.
1. Harmonious Passion	5.07	1.14	-	0.54 ***	0.69 ***	0.47 ***	0.25 **	0.35 **	0.24
2. Obsessive Passion	3.85	1.73	0.47 **	-	0.19 **	0.63 ***	0.60 ***	−0.12	0.58 ***
3. Mastery Goals	5.32	1.14	0.62 **	0.17 **	-	0.51 ***	0.28 **	0.50 ***	0.02
4. Performance-Approach Goals	4.64	1.36	0.42 **	0.57 **	0.47 **	-	0.77 ***	0.08	0.50 *
5. Performance-Avoidance Goals	4.45	1.40	0.21 **	0.54 **	0.25 **	0.69 **	-	0.03	0.52 **
6. Team-Oriented Decisions	8.25	1.76	0.30 **	−0.10	0.42 **	0.10	0.05	-	−0.06
7. Self-Oriented Decisions	6.51	1.92	0.34 **	0.52 **	0.20 **	0.53 **	0.48 **	0.09	-

Note. *N* = 253; *M* = mean; *SD* = standard deviation; correlations based on scale scores are reported below the diagonal; latent variable correlations from the CFA measurement model are reported above the diagonal; * *p* < 0.05; ** *p* < 0.01; *** *p* < 0.001.

## Data Availability

The raw data supporting the conclusion of this article is available upon request from the corresponding author.

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
