# Peer review of "The Role of Passion in Self-Oriented Versus Team-Oriented Decision-Making in Team Sports"

_ijerph, 2023, doi:10.3390/ijerph20032626_

Round 1
Reviewer 1 Report
The article is a study that investigates the role of passion and achievement goals in self-oriented and team-oriented decision making in basquetball. It is a research work of good quality in terms of justification of the subject matter, the method used, the results and the discussion. The only suggestion is to deepen in the conclusions, by virtue of the proposed objetives and the findings evidenced in basketball players, in order to establish projections for future research.
Author Response
Response to Reviewer 1’s Comments
“The article is a study that investigates the role of passion and achievement goals in self-oriented and team-oriented decision making in basquetball. It is a research work of good quality in terms of justification of the subject matter, the method used, the results and the discussion.”
Point 1: “The only suggestion is to deepen in the conclusions, by virtue of the proposed objectives and the findings evidenced in basketball players, in order to establish projections for future research.”
Response 1: First, thank you for the positive feedback. It is highly appreciated. Second, we agree that the conclusions should be deepen. Thus, changes were made throughout the discussion accordingly.
Reviewer 2 Report
I begin by thanking the editor for the opportunity to review this work.
I congratulate the authors for the work done and for chosen of theme.
The theme of this study is very interesting and statistical data are important.
In general, the research was well built, and the results they have achieved are exciting, but, the article would benefit from some improvements.
First, 64% of participants are no longer athletes. Could this fact not bias your perception and consequently the results? Authors performed a comparative statistical test between athlete’s results and former athletes? These results should be present.
The authors present two empirical hypotheses, but they are not in evidence on text. The paper it would benefit for these hypotheses to become evident and for them to be accompanied by a justification or theoretical argument.
Finally, the discussion should present the theoretical and mainly practical implications of the findings. The discussion only reflects the results, there was no real reflection on them, this diminishes the quality and relevance of the work performed.
Kind regards
Author Response
Response to Reviewer 2’s Comments
Point 1: “First, 64% of participants are no longer athletes. Could this fact not bias your perception and consequently the results? Authors performed a comparative statistical test between athlete’s results and former athletes? These results should be present.”
Response 1: As mentioned in section 2.1. Participants and Procedure, all participants were still playing basketball when they completed the questionnaire. Thus, they were still athletes at the time of the study. However, we understand that the wording of the sentences (lines 99 to 102 and 102 to 105) may have been confusing. Accordingly, we have made changes as follows: “To participate in this study, participants had to have played basketball in an official team sanctioned by a league either at a high school, college, or professional level and to still be playing basketball at the time of the study, either recreationally or in an organized league.” and “Most participants played organized basketball in high school (64.0%), while others played either at a college (36.0 %) or professional (2.8%) level, for an average of 8.0 (SD = 6.4) years of organized basketball.”
Point 2: “The authors present two empirical hypotheses, but they are not in evidence on text. The paper it would benefit for these hypotheses to become evident and for them to be accompanied by a justification or theoretical argument.”
Response 2: In line with the reviewer’s comment, we now use the word “hypotheses” and thus underscore the hypotheses by rephrasing the sentence where they were first presented (line 82) as follows: “In line with the above, it was hypothesized that HP mostly predicts mastery goals, whereas OP predicts all three types of goals but mostly performance-approach and performance-avoidance goals.”
Point 3: “Finally, the discussion should present the theoretical and mainly practical implications of the findings. The discussion only reflects the results, there was no real reflection on them, this diminishes the quality and relevance of the work performed.”
Response 3: We agree with the Reviewer. Thus, changes were made in the Discussion section accordingly. The most important change pertains to a whole paragraph that was written on practical implications (p. 6, lines 260 to 275). It reads as follows: “The present findings lead to many practical applications. Indeed, results highlight the importance of fostering HP in sport and how it can benefit a team-oriented decision. First, HP can be fostered through the autonomy support that coaches offer to athletes. The DMP states that autonomy support facilitates the development and maintenance of HP whereas controlling behaviors facilitate OP [1]. Coaches can support athletes’ autonomy in many ways, such as offering choices, providing task rationale, avoiding punishment, providing non-controlling feedback, and recognizing athletes’ feelings and perspectives [38]. Second, harmonious engagement in sport can also be facilitated by promoting athletes’ life balance. Balancing sport with other aspects of their lives (e.g., school, work, family) can allow athletes to flexibly engage in their sport, ultimately leading to positive consequences not only in sport, but also in other aspects of their life [1]. Third, it has been shown that increasing the use of signature strengths is related to higher levels of HP and well-being [39]. Thus, coaches could guide athletes in the discovery of their own athletic strengths and lead them to use these strengths in sport. In doing so, athletes should feel that they are using their full potential and engage more harmoniously in their sport, thereby leading to team-oriented decisions.”